# Increased signal-to-noise ratios within experimental field trials by regressing spatially distributed soil properties as principal components

Jeffrey C Berry[1], Mingsheng Qi[1], Balasaheb V Sonawane[2], Amy Sheflin[3], Asaph Cousins[2], Jessica Prenni[3], Daniel P Schachtman[4], Peng Liu[5], Rebecca S Bart[1]*

[1]Donald Danforth Plant Science Center, St. Louis, United States; [2]School of Biological Sciences, Washington State University, Pullman, United States; [3]Department of Biochemistry and Molecular Biology, Colorado State University, Fort Collins, United States; [4]Department of Agronomy and Horticulture, University of Nebraska-Lincoln, Lincoln, United States; [5]Department of Statistics, Iowa State University, Ames, United States

**Abstract** Environmental variability poses a major challenge to any field study. Researchers attempt to mitigate this challenge through replication. Thus, the ability to detect experimental signals is determined by the degree of replication and the amount of environmental variation, noise, within the experimental system. A major source of noise in field studies comes from the natural heterogeneity of soil properties which create microtreatments throughout the field. In addition, the variation within different soil properties is often nonrandomly distributed across a field. We explore this challenge through a sorghum field trial dataset with accompanying plant, microbiome, and soil property data. Diverse sorghum genotypes and two watering regimes were applied in a split-plot design. We describe a process of identifying, estimating, and controlling for the effects of spatially distributed soil properties on plant traits and microbial communities using minimal degrees of freedom. Importantly, this process provides a method with which sources of environmental variation in field data can be identified and adjusted, improving our ability to resolve effects of interest and to quantify subtle phenotypes.

*For correspondence:
rbart@danforthcenter.org

## Editor's evaluation

In this study, the authors took an experimental, empirical approach to tackle the thorny problem of micro-scale variation in soil properties within and among field plots in confounding statistical analyses. The issue is that in field experiments, small variation in one or more soil property variables can obscure true effects of experimental variables on plant phenotypes. The main result is that without their framework they would not have found the association between water treatment, plant growth and Microvirga bacterial abundance, it would have been lost to the noise inherent in these kind of large-scale experiments with relatively modest degrees of freedom. Overall, the PC-based approach to de-noise these kinds of datasets provides an important advance by pulling out subtle phenotypic effects in field trials.

## Introduction

Environmental variation makes the real world a noisy place to conduct science. In the context of experimental agriculture fields, variation in topography may result in uneven water moisture accumulation. Similarly, soil nutrients such as nitrogen and phosphate are often nonuniformly distributed

across a field. These unintended and often unknown sources of environmental variation may significantly affect the experimental results. The traditional approach to mitigate this variability is through experimental designs that include replicate blocks (*Piepho et al., 2013*; *Fisher, 1925*). While helpful for controlling for variation that is relatively uniform within the blocks, true biological signal may still be masked by other experimental noise that is heterogeneous within blocks.

Analytical approaches have been used to parameterize spatial variation within a field using traditional mixed-effect modeling. These methods come in two general flavors: estimating spatial-covariance structures and spatial-smoothing using splines (*Rodríguez-Álvarez et al., 2016*; *Piepho et al., 2008*; *Velazco et al., 2017*). The former is older and canonical but requires advanced statistical knowledge to interpret the results. The latter is newer and easier to use courtesy of advancements in computation. Spatial-smoothing has been shown to effectively account for spatial variations in uniform barley fields and promotes genetic heritability in simulation studies (*Rodríguez-Álvarez et al., 2016*). While spatial-smoothing using splines does effectively address spatial variation of a trait in a field, traditional parameterizations using spatial-covariance structures do so as well and further provide intuitive metrics on the type and shape of the structure. Estimating and accounting for spatial structure have proven useful for a variety of biological systems including nematodes (*Quist et al., 2019*), microbiomes (*Franklin and Mills, 2003*), forestry (*Ohashi and Gyokusen, 2007*; *Möttönen et al., 1999*; *Bai et al., 2012*), and ionomics QTL mapping (*Pauli et al., 2018*). These previous studies have used spatial-covariance estimation methods to identify and associate spatial effects on various traits of interest. These methods are also the backbone of geospatial statistics where the goal is to interpolate values between sampling points (*Olea, 2018*).

Another challenge facing field studies is to identify which factors of a multivariate dataset influence the traits of interest and then adjust for the effects from all covariates while maintaining sufficient degrees of freedom for statistical inference. This challenge is similar to the challenge associated with genome-wide association studies (GWAS) that must handle population structure. In GWAS studies, phylogenetic relatedness is managed by principal component analysis (*Price et al., 2006*). Principal components (PCs) capture axes of most variation and effectively reduce a complex multivariate dataset down to only a few independent vectors of greatest importance. PCs have also previously been used for dimension reduction to investigate how various soil nutrients affect natural selection on plant roots (*Murren et al., 2020*).

Here, we combine approaches from geospatial statistics with dimension reduction to deal with environmental variation in field studies. Specifically, we estimated spatial-covariance structures for each factor and then accounted for these effects using principal component regression. Applying this approach reduced experimental noise due to environmental variation and revealed previously hidden associations in a field study. This field study with 24 varieties of sorghum and 2 watering treatments, well-watered and water stressed, arranged in a split-plot design with 8 replicate blocks, was completed in 2017. Several types of data were collected including, but not limited to, plant harvest traits (height, fresh and dry weight, and panicle size), soil property composition (calcium, magnesium, nitrate, organic matter, pH, phosphate, potassium, salinity, sodium, sulfate, and total cations), three microbiome samplings for each plot (root, rhizosphere, and soil), leaf traits (specific leaf area, C and N content, and stable isotopes of C and N), and root metabolomic profile. In this study, the soil chemical and physical properties were used as the multicovariates that exhibited spatial-covariance structure and subsequently created microtreatment effects throughout the field that are associated with plant traits. We demonstrate that accounting for these effects via residuals of principal component regression is an effective method to improve the detection of effects of treatment factors and reduce the noise caused by spatial variation within a field.

## Results

### Geospatial statistics interpolates soil property composition throughout the field

We previously described a field-level experiment in which sorghum and its associated microbiome were evaluated across two different watering treatments (*Qi et al., 2022*). As is typical of field studies, the collected data showed significant variability across all measured parameters. We also measured several different soil properties at multiple points throughout the field including organic matter, pH,

phosphate, nitrate, sum of cations, calcium, magnesium, potassium, sodium, sulfate, and salinity compositions. We hypothesized that variation across the field in these soil properties may explain some of the variability in the other measurements. Here, using the field and microbiome datasets from *Qi et al., 2022*, we describe and evaluate a general method for reducing noise and apply this method to additional metabolomics and stable isotope datasets.

First, because only a limited set of points across the field were sampled for soil properties, there was a need to estimate the missing values (*Figure 1A*). We suspected that sample proximity would introduce correlation within the measured soil properties. To assess this type of spatial correlation, we employed techniques from geospatial statistics to capture the correlation structure of any pair of samples in the field. Of the 12 properties tested, 6 properties exhibited evidence of spatial distributions ($p$ value $<0.05$: salinity (mmho), nitrate (ppm), sulfate (ppm), calcium (ppm), magnesium (ppm), and phosphate (ppm) (see Methods: Statistical testing for evidence of spatial structure). For these six soil properties we estimated the missing values throughout the field. Interpolation of values between sampling points was performed by leveraging spatial correlation structure to predict unobserved values, a process called kriging (see Methods: Geospatial interpolation methods) (*Table 1*). To test the kriging accuracy, we performed a leave-one-out cross-validation for each soil property. Through this analysis we observed that the error of the predictions, when scaled to unit variance of the observations, exhibit distributions that resemble the expected standard Z-distribution (*Figure 1—figure supplement 1*). The ratio of the partial sill to the nugget is a proxy for the magnitude of the variance that is attributable to the spatial structure. Comparing this ratio for nitrate and phosphate shows that the phosphate spatial correlation was much stronger than the nitrate spatial correlation (*Figure 1*). Calcium also exhibited correlation structure of distances larger than nitrate, but much smaller than phosphate (*Figure 1*). To visualize the spatial structure for each soil property, the kriged values of each property were centered around the mean and scaled to unit variance (*Figure 1*). This analysis revealed that the soil properties exhibited different topographies across the field. For example, phosphate levels were high in a band across the center of the field while nitrate and calcium levels were more variable with several high and low spots (*Figure 1*). We also considered correlation between the different soil properties and observed several correlation blocks, implying similar spatial structures (*Figure 1—figure supplement 2*).

## Soil property variation influences plant phenotypes and microbiome composition

The above analyses clearly showed that soil properties were variable across the field site. However, it was not clear whether the observed variation was large enough to affect plant-associated phenotypes or the microbiome. To address this, we used constrained analysis of principal coordinates (CAP). With CAP, it is possible to identify specific effects on a multidimensional dataset while acknowledging variation due to other effects. For example, to understand if and how soil property variation affected microbiome composition, we first had to control for the effects of the watering treatments, the different genotypes and their interactions (see Methods: Statistical testing for phenotype–property associations). First, CAP was computed and a permutation analysis of variance (ANOVA), using 999 iterations, was performed on each soil property to identify if there is an interaction with the different compartments on the overall microbiome composition and resulted in all but one soil property, sulfate, showing an interaction effect (*Table 2*). Following the interaction effect permutation ANOVA, post hoc permutation ANOVAs, using 999 permutations, were performed to identify specific soil property influence on the three compartments independently. From this analysis, we observed that the root microbiome was invariant to all soil property variations. In contrast, the rhizosphere and soil microbiomes were influenced by the variation in several soil properties (*Figure 2A*). Additionally, there were some soil properties (salinity, sulfate, and calcium) whose variation affected either the rhizosphere or soil microbiomes but not both. This suggests that microbiome compartments are differentially sensitive to different types of soil property variation. CAP and permutation ANOVA were also applied to annotated root metabolomic profiles. In contrast to the large effects seen in the microbiomes, no soil property variances were associated with changes in the metabolomic profile (*Figure 2—figure supplement 1*). This may reflect the relative stability of the metabolites identified from gas chromatography–mass spectrometry (GC–MS) which mostly represents primary metabolites, or the sensitivity of measurements.

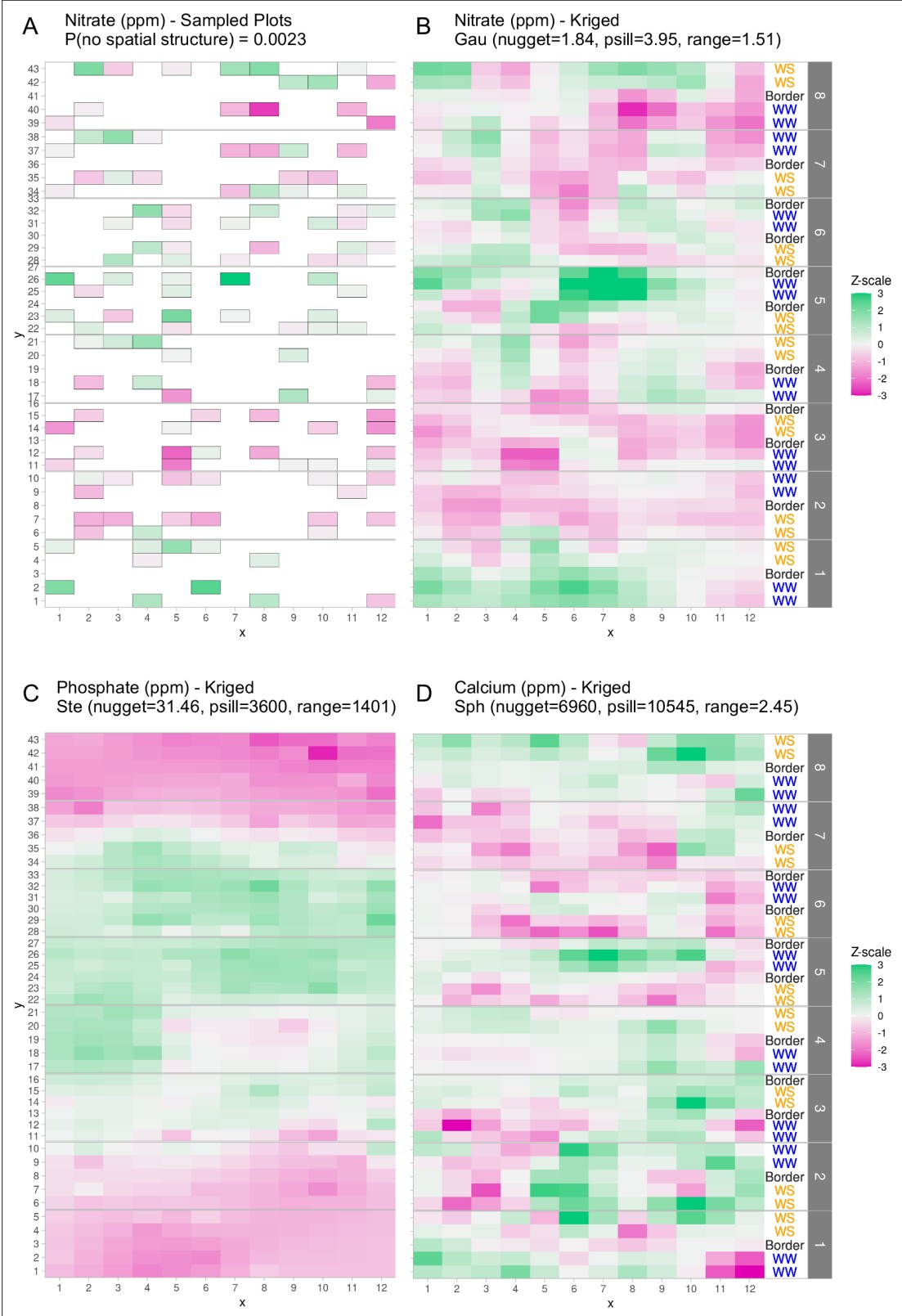

**Figure 1.** Graphical depictions of field layout where each cell is a plot in the field. Water treatment is specified on right; WS = water stressed, WW = well-watered. Eight split-plot replicate blocks are denoted in gray vertical bars. Color scale represents data with genotype and treatment removed. Green indicates larger than average, white indicates approximately average, and magenta indicates below average values. (**A**) Nitrate values are shown

*Figure 1 continued on next page*

*Figure 1 continued*

for each cell (outlined in gray) that were sampled for soil property analysis. (**B**) Kriged nitrate values to estimate nitrate levels in unsampled plots. (**C, D**) Kriged values for phosphate and calcium. Variogram fit of spatial model is indicated with model type, nugget, partial sill, and range.

The online version of this article includes the following figure supplement(s) for figure 1:

**Figure supplement 1.** Leave-one-out cross-validation of soil property observations.

**Figure supplement 2.** Pearson correlations between all pairwise soil properties.

While microbiome and metabolite data are highly multivariate, plant phenotypes, composed of individual phenotypic traits, are much less so and are therefore suited to univariate statistical analyses. Just as the microbiome has the potential to be influenced by the soil property variations, the same could be true for the univariate phenotypic traits. Similarly, the effects due to the experimental design must be acknowledged to more precisely evaluate the association a given property has on the phenotypic trait. Mixed-effect models were created for each soil property–phenotypic trait pair with treatment, genotype, and their interaction as fixed effects and random effect for the split-plot replicates each having multivariate-normal spatial correlation structure (see Methods: Statistical testing for phenotype–property associations; *Equation 4*). The precise effect a given soil property has on a phenotypic trait was evaluated using type III sum of squares to account for the other sources of variance (treatment, genotype, and the interaction) on the phenotypic trait. Traditional *F*-test from the ANOVA revealed that plant height and total fresh weight are influenced by soil phosphate and magnesium variation, respectively (*Figure 2B*). Similar modeling of the leaf traits indicated the soil phosphate variation is significantly associated with $\delta^{15}$N (*Figure 2C*). Many other phenotypic traits were examined and did not have statistically significant associations with the variation in soil properties (*Figure 2—figure supplement 1*). Closer examination showed that soil phosphate levels are mildly inversely correlated to plant height (*Figure 2D*). This supports the hypothesis that excess phosphate inhibits plant growth and development (*Shukla et al., 2017*; *Song et al., 2016*) and suggests that the levels found in the center of the field were too high for optimum sorghum growth.

## Statistical approach to place field noise into PCs

We have shown that many of the soil properties exhibit spatial distribution and influence various plant and microbiome traits. Therefore, to understand the effects of treatment and genotype on the phenotypic traits more precisely, we must first account for the effects of the soil properties. The replication in our study was not sufficient to include all the soil properties as covariates to account for their influence – this would require a degree of freedom for every soil property. A generalized approach to overcome limited sample size is reducing the dimensionality of the covariates using principal component analysis and regressing against the first several PCs, known as principal component regression. In this approach, the PCs retain a percentage of the influence from the individual properties and can be used as a proxy to adjust for as much variation as possible.

To adjust for the spatial effect of soil properties unrelated to genotype and treatment, we used the observed soil properties, not the kriged values, and fit linear models for each soil property with

**Table 1.** Shown are sum-of-square errors for each soil property (columns) and each spatial model tested (rows).
Cells that have asterisks are those models that have minimal errors and are chosen to be the best-fit model for kriging for the respective soil property.

|  | Salinity | Nitrate | Sulfate | Calcium | Magnesium | Phosphate |
|---|---|---|---|---|---|---|
| Nugget only | 1.92115E−05 | 203.668 | 10645.6 | 833211000 | 2261360 | 149,688 |
| Exponential | 1.1769E−06 | 41.2651 | 1909.53 | 17886300000 | 65136300 | 27556.2 |
| Spherical | 2.03099E−06 | 25.5247 | 2058.42 | 244155000* | 1038010* | 32351.9 |
| Gaussian | 2.16269E−06 | 22.3183* | 289,060 | 17669600000 | 64365000 | 100,531 |
| Matern | 0.00000113273* | 23.0069 | 1682.03* | 266373000 | 1087980 | 18961.6 |
| Stein's Matern | 1.13273E−06 | 23.0069 | 290,357 | 17886300000 | 65136300 | 18827.8* |

**Table 2.** Shown for each soil property that exhibits significant spatial distribution (see Methods: Statistical testing for evidence of spatial structure) (first column) is the interaction effect with the different microbiome tissue compartments on the overall microbiome composition (p value <0.05 indicates significant interaction, PERMANOVA from model: Composition ~ Property:Compartment).

| Soil property | p value |
|---|---|
| Sulfate (ppm) | 0.24 |
| Salinity (mmho/cm) | 0.011 |
| Phosphate (ppm) | 0.001 |
| Nitrate (ppm) | 0.001 |
| Magnesium (ppm) | 0.003 |
| Calcium (ppm) | 0.005 |

fixed effects being treatment, genotype, and their interaction. The residuals from these linear models were the adjusted soil properties unrelated to genotype and treatment. These residuals were then processed with principal component analysis. To test the success of accounting for our experimental design, we investigated clustering within the first two PCs and observed that indeed the treatment and genotype effects were accounted for (*Figure 3A*) as there were no obvious treatment or genotype effects remaining in these PCs. We selected the first three PCs as regressors which represented approximately 66% of the total variance in the soil property data (*Figure 3B*). Next, we visualized the contribution of each soil property within each PC (*Figure 3C*). PC1 is primarily represented by calcium, magnesium, potassium, sodium, and sum of cations. PC2 is primarily represented by sulfate, salinity, and pH. PC3 is primarily represented by phosphate and nitrate. Then, we used kriging (see Methods: Geospatial interpolation methods) to interpolate the missing values for the rest of the field. Since many soil properties exhibited spatial distributions (*Figure 1*), we expected that the PCs would also display a spatial distribution. Indeed, the spatial distribution of the kriged first PC resembles the calcium distribution which emphasizes the contribution of that property (*Figures 1 and 3D*). In summary, through this approach, we revealed specific

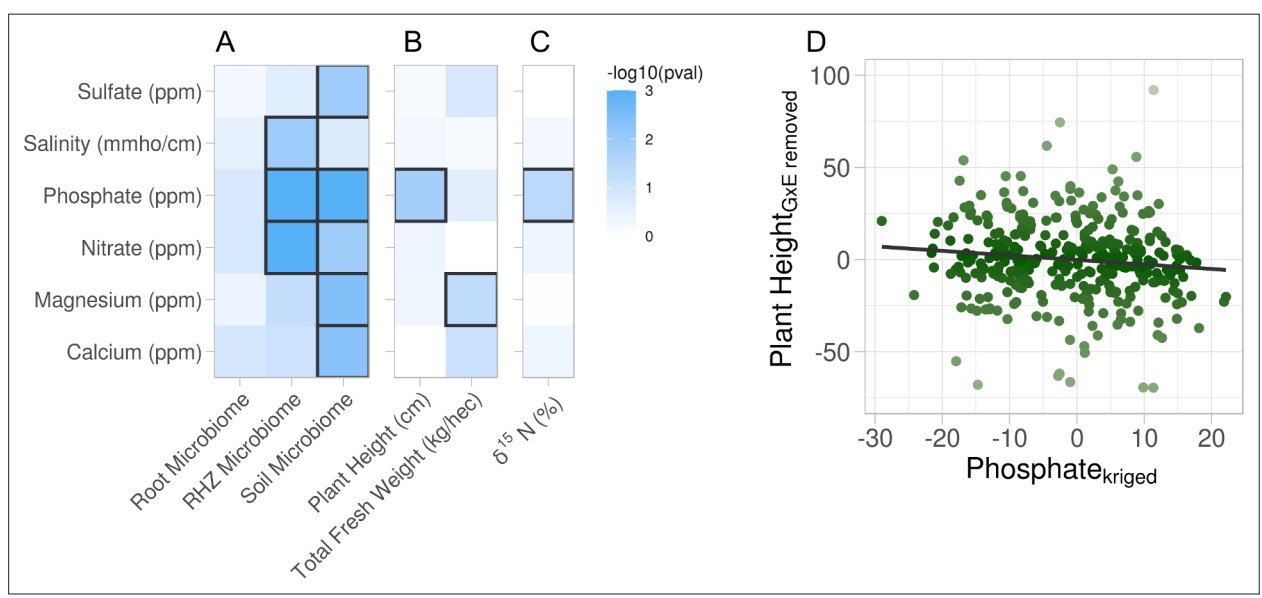

**Figure 2.** Association of soil property variations with multiple phenotypes. Six soil properties were assessed for effect on root microbiome and plant phenotypes using permutation ANOVA. Cells are colored by −log10 p value of the effect. (**A**) Effect of each soil property on microbiome beta diversity from three root compartments: root (endosphere), RHZ (rhizosphere), and soil (bulk soil), while constraining on genotype and treatment. Effect of each soil nutrient on the height and weight (**B**) and leaf $\delta^{15}N$ (**C**) using type III sum of squares while including treatment, genotype, and interaction as additional fixed effects. (**D**) Example effect of kriged phosphate, x-axis, on plant height, adjusted for genotype and treatment, y-axis.

The online version of this article includes the following figure supplement(s) for figure 2:

**Figure supplement 1.** Same analysis as *Figure 2A-C*, but here are shown the phenotypes that did not demonstrate soil property associations.

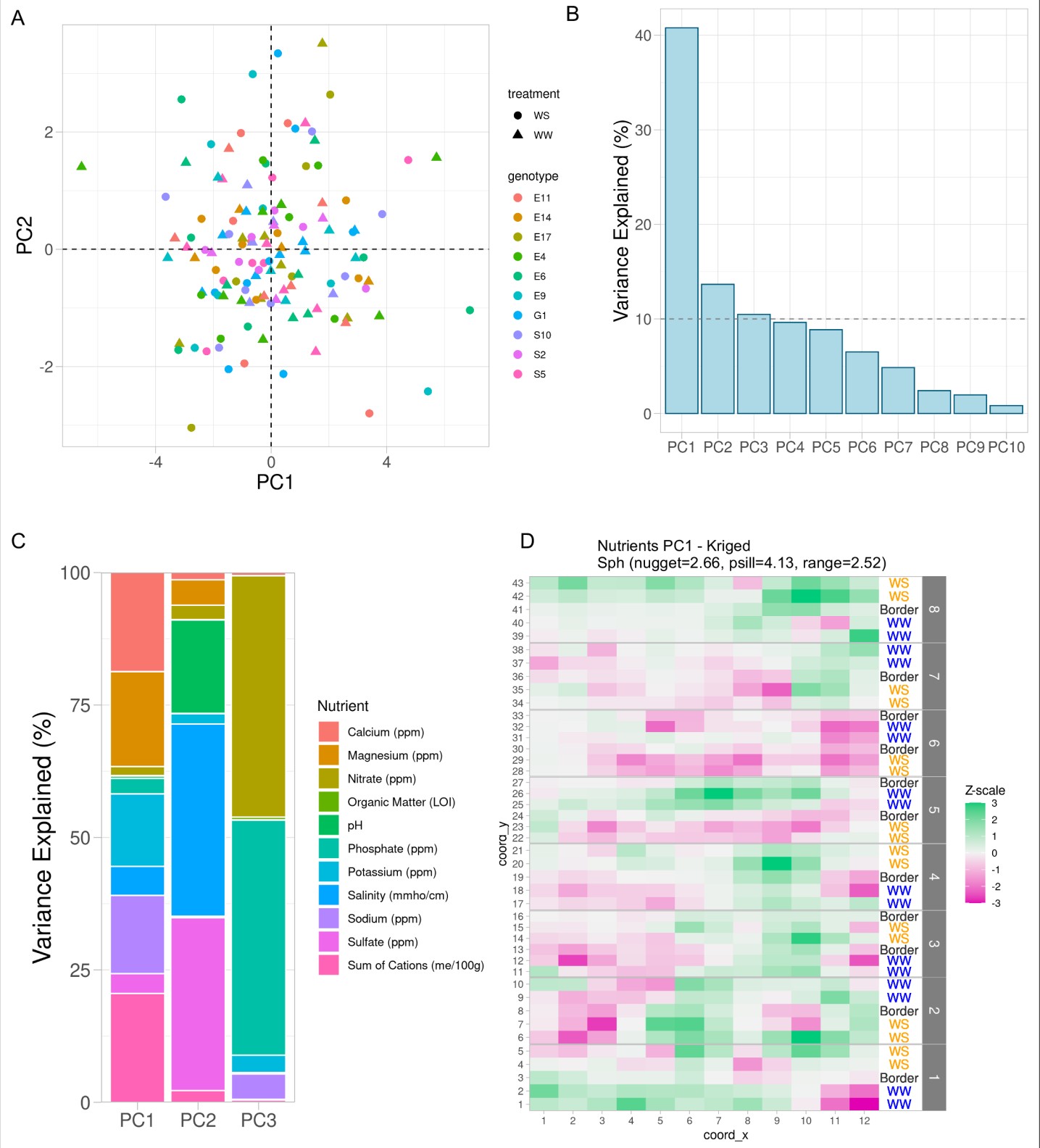

**Figure 3.** Variation in soil properties can be collapsed into principal components. (**A**) First two principal components of soil property residuals as *x*- and *y*-axis, respectively, colored by genotype and shaped by treatment. (**B**) Scree plot of the first 10 principal components. Shown is the percent variance explained of the total property variance by each component. Dashed line is at 10% variance explained. (**C**) For the first three components, colored is the contribution of each soil property to its respective variance explained within each component. (**D**) Spatial distribution of kriged PC1. Each cell colored by scaling the values to unit variance. Variogram fit with nugget, partial sill, and range displayed.

sources of field-level noise and successfully captured a significant proportion of the field-level variation into three PCs.

## Using PCs to denoise field data

Having reduced the soil property influences into a limited number of PCs, next we aimed to account for these unintended influences on our phenotypes of interest. Similar to how we removed the experimental design effects, we created mixed-effect models for each of our phenotypes as the dependent variable with multivariate normal spatial structure within each split-plot replicate, and the first three PCs as fixed effect independent variables (see Methods: Principal component regression; *Equation 5*). For the univariate phenotypes, we extracted model residuals (*Equation 2*) to generate soil property invariant phenotypes. In the microbiomes, residuals were not comparable to the original counts, therefore we back transformed the model estimates to create an adjusted count for each operational taxonomic unit (OTU) (see Methods: Principal component regression; *Equation 6*).

In the microbiome datasets, we expected the variance within each of the compartments to decrease and the difference between compartments to be more obvious. To test this, we combined the original observed counts and the adjusted counts for microbes that were able to be modeled and performed principal component analysis. We observed larger distances between microbiome compartments using the adjusted counts versus the observed counts (*Figure 4A*). This indicates the sources of variation from the soil properties were better controlled thereby increasing the differentiation between compartments. Within each compartment, the root microbiome was least affected by soil factors which was demonstrated by the small distance between the observed and adjusted counts, followed by rhizosphere with a larger separation, and soil being the farthest and most affected (*Figure 4A*). The adjusted counts produced clusters that were larger than their respective original counts, again indicating sources of variation were addressed so that the within compartment effects were better elucidated (*Figure 4A*). Next, we examined the experimental design effects within each compartment using variance components (PERMANOVA) and found mirroring results (*Figure 4B*). The experimental design was not differentially resolved in the root microbiome after principal component regression; however, the rhizosphere and soil microbiomes saw an increase in 3% and 2%, respectively. Noting that treatment was the effect of largest change, we sought to visualize treatment effect changes within each of the tissue compartments. Given the strong compartment differentiation, identifying clusters within a compartment required principal component analysis to be performed on each compartment individually. In the rhizosphere, we observed treatment differentiation using both the observed and adjusted counts; however, the distance between the centers of each cluster was larger after adjustment further indicating within-group variation was reduced (*Figure 4—figure supplement 1*).

To assess how principal component regression affected the univariate plant phenotypes, we compared the data before and after principal component regression. Prior to removing noise from soil properties, plant height and fresh weight both showed drought effects, and after performing principal component regression, these effects were maintained (*Figure 4—figure supplement 2*). Further, by plotting the original values against the change for each value after adjustment, we showed that this correction method is equitable for all plant sizes; in other words, short and tall plants were not overly adjusted either positively or negatively (*Figure 4*). Additionally, the variation attributable to soil properties was as much as ±10% for fresh weight and ±6% for plant height for some individual observations. Additionally, variance components were computed for all harvest traits but both fresh weight and plant height did not demonstrate an increase in experimental design resolution (*Figure 4—figure supplement 2*). Lastly, leaf delta $\delta^{15}N$ showed an association with phosphate (*Figure 2C*), and variance components for the experimental design effects found that the $R^2$ between the unadjusted and adjusted increased by approximately 5% (*Figure 4E*). This is similar to the changes in the explained variance in rhizosphere and soil microbiome compositions.

Microbiome analyses often have less-than-ideal replication and therefore elimination of confounding variations is crucial in identifying effects of interest, such as identifying plant-growth promoting microbes. For instance, change-point models can identify microbial impacts on plant phenotypes once the abundance of the microbe reaches a particular level (*Qi et al., 2022*). We show that in water-stressed samples before the adjustment on both plant height and microbe abundance of *Microvirga* in the soil microbiome, we do not see any evidence of an association. However, after accounting for the variance from the soil properties, there is a strong positive correlation between plant phenotype

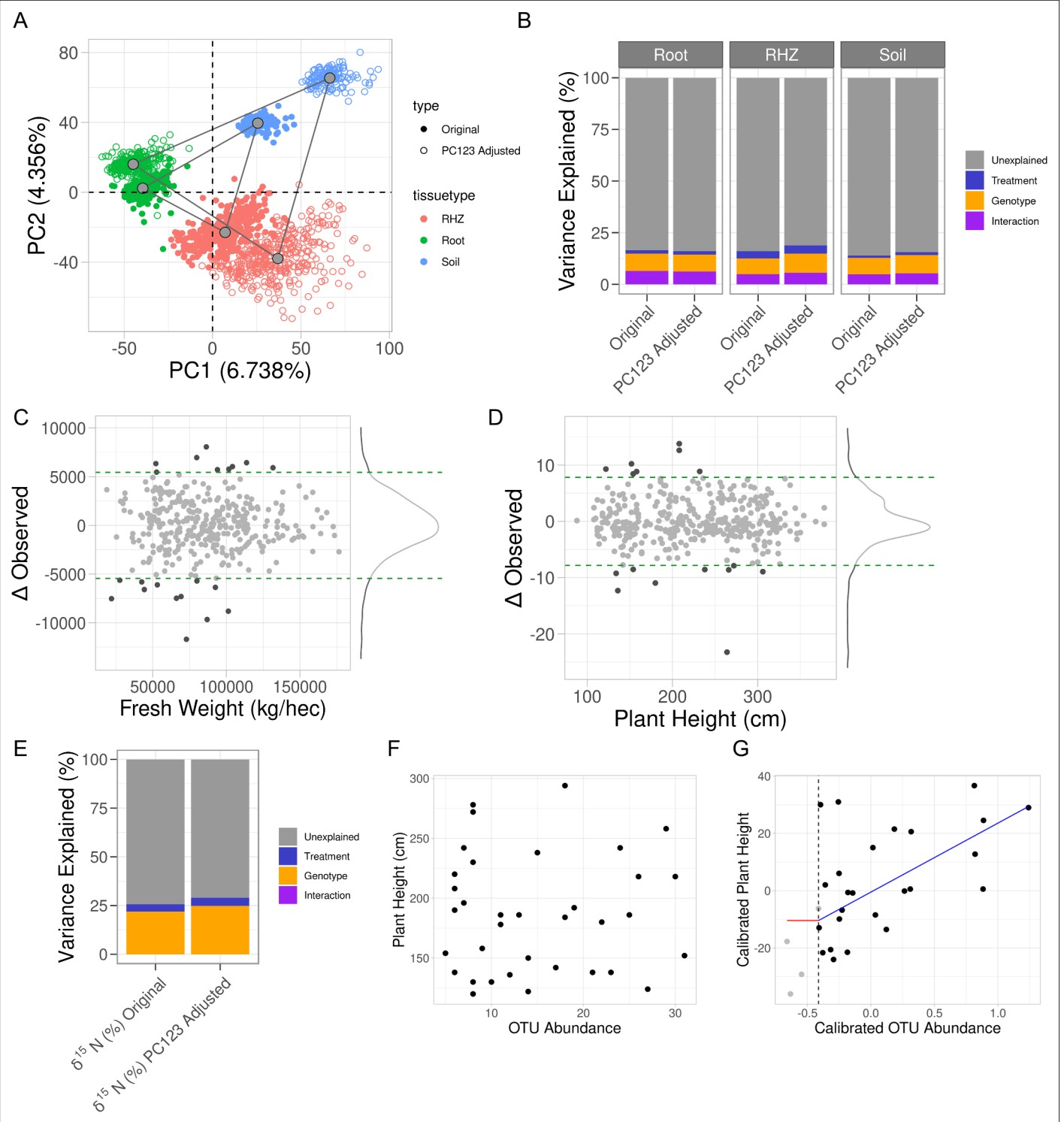

**Figure 4.** Accounting for influence from soil property variance within microbiome data reveals plant phenotypes that correlate with microbe abundance. (**A**) Principal component analysis on the combined raw and residual microbiome tables. Shown are the first two components with their respective variance explained. Samples are colored by tissue type and shaped by original or residual values. Gray points are the centers of each respective cluster, and gray lines connect the centers of each cluster. (**B**) Partial correlations of experimental design variables in each microbiome compartment's composition before and after principal component regression. (**C**) Observed plant height values, *x*-axis, and the change in that value as a result of the adjustment, *y*-axis. (**D**) Similar to C, shown are the fresh weight values and their respective changes. In both C and D, horizontal green dashed lines represent the 95% confidence interval for the change in observation. Light gray dots are within the interval, and dark gray dots are outside of the interval. (**E**) Partial correlations of experimental design variables in leaf δ15N before and after principal component regression. (**F, G**) For only water-

*Figure 4 continued on next page*

*Figure 4 continued*

stressed samples and only the soil microbiome, plant height, *y*-axis, and operational taxonomic unit (OTU) abundance of Microvirga, *x*-axis, before and after principal component regression. Shown in G, is the fit of a change-point model where the red line is no change before threshold, the vertical dashed line, and the blue line is a linear fit after the threshold.

The online version of this article includes the following figure supplement(s) for figure 4:

**Figure supplement 1.** Principal component analysis on the root (**A**), rhizosphere (**B**), and soil (**C**) samples using the combined raw and residual microbiome tables.

**Figure supplement 2.** Effect of applying principal component regression on plant morphology.

and OTU abundance (false discovery rate (FDR)-adjusted p value <0.05) (*Figure 4F, G*). This further emphasizes that if an experimental factor is expected to have a small effect size, that it would be lost to noise. We observe an order of magnitude more significant OTUs in the change-point associations before principal component regression versus after (*Table 3*). Adjusting the soil property effect or other nuisance factors in experimental design can lead to increased power of detecting biological signals.

## Discussion

Large-scale trials within complex environments are an important component of many biological subdisciplines. Because of environmental variability, these experiments must include high levels of replication and even so, results often fail to repeat in subsequent trials. For agricultural field studies a major source of variation is heterogeneous soil property distributions that create their own microtreatments and are covariates to planned experimental designs. Because these microtreatments are often unknown, and therefore not accounted for, they show up as experimental noise and may lead to false inferences. For instance, nitrogen is known to affect plant growth (size, color, yield, etc.) (*Veley et al., 2017*; *Chapin, 1987*). If nitrogen is unevenly distributed across a field experiment aimed at characterizing biomass among diverse genotypes, the variability in nitrogen may confound the experiment. In this study, by intentionally measuring multiple soil properties across the field experiment, we were able to account for this known variation through PCs and gain novel biological insight into possible field relevant interactions between plants and microbes (*Qi et al., 2022*).

After accounting for environmental variation, we observe a reduced total number of OTUs that correlate with plant phenotypes (*Table 3*). While the total number is smaller, we also observe new 'hits' that were not observed before accounting for environmental factors. Additional work will be required to fully understand these patterns but here we offer a few theoretical explanations. We note that accounting for the environmental variation reduced our degrees of freedom that may reduce the power of detecting some associations. More importantly, after adjusting for environmental effect, we have focused our analysis on conditional associations instead of marginal associations. *Marginal associations* measure whether two variables change in the same or opposite directions simultaneously, regardless of other factors that may affect those two variables. *Conditional associations*, on

**Table 3.** Shown for each microbiome compartment are the number of operational taxonomic units (OTUs) that showed significant association (p value <0.05) in their abundance to the respective phenotype, either positive or negative, both before (original) and after (PC123 Adj) principal component regression.

The intersection column shows the number of OTUs shared between these two sets of counts.

| Compartment | Phenotype | Original | PC123 Adj | Intersection |
|---|---|---|---|---|
| Soil | Dry weight | 7950 | 445 | 155 |
| RHZ | Dry weight | 5619 | 897 | 333 |
| Root | Dry weight | 3320 | 231 | 96 |
| Soil | Plant height | 7991 | 342 | 137 |
| RHZ | Plant height | 5614 | 854 | 329 |
| Root | Plant height | 3397 | 245 | 97 |

the other hand, adjust for the impacts of other variables, and measure the relationship between two variables that cannot be explained by other variables. In a scenario in which the environment affects a microbe abundance and a plant phenotype, this may be observed as marginal associations between the two even though the microbe abundance and plant phenotype are not directly associated with each other. It is possible that many of the 'hits' before environmental factor adjustment are results of such marginal associations. By accounting for environmental factors, we focus on conditional associations that measure the rarer direct or possibly causal interactions between OTUs and plants. Nevertheless, future experimental work will be required to refine the list of candidate microbes.

While the approach described here represents a major advance forward, we acknowledge several opportunities for further improvement. For example, the soil property data were collected at just one time point, approximately 1 month prior to harvest. While we expect some soil properties to remain constant over the course of the experiment, others likely fluctuate (e.g., water content). Future studies might gather soil property composition at multiple time points, including before planting and at the time of phenotyping, to generate paired data. Additionally, advancements have been made in spatiotemporal modeling using Bayesian hierarchical modeling with time as an autoregressor (*Finley et al., 2015*; *Rushworth et al., 2014*) which may prove powerful if soil property composition were densely sampled over time. We also note that replication remains a crucial aspect of these types of experiments. In this analysis, we used three degrees of freedom by including the first three PCs of the soil property data in a regression to account for their contributions on the phenotypes. We chose to only include PCs that have at least 10% variance explained and only include up to replication number minus three so the experimental design effects could still be estimated. In our case, had a fourth PC shown a significant source of variance, degrees of freedom would have become limiting. On the other hand, had we included additional replication, it may have been possible to correct for covariates such as the soil properties by directly regressing on the properties themselves, rather than using a dimension reduction technique.

A significant advantage of the method described here, is that the sources of variation (soil properties) are known. This allows the researcher to examine how these fluctuations might influence experimental results and may also lead to additional biological insight. However, we note that it is not possible to measure every potential source of variation. In many cases, it may prove useful to apply an approach that is agnostic to the source of variation, such as the SpATS method described previously (*Velazco et al., 2017*).

We note that in these datasets, the stable isotopes and primary metabolite profiles are invariant to the measured soil properties. For the isotopes, this may be indicative of stability relative to the fluctuations in the properties across the field, but it is possible that if the soil property variations were larger, then a relationship might be established. The metabolite profile used in these analyses is only those captured from GC/MS and mostly consist of primary metabolites such as sugars, organic and amino acids, small phenolics, and fatty acids. It may be true that secondary metabolites that were not examined in these analyses may associate with the soil property variations. We observe that a relatively small amount of variation in plant height and weight were attributed to the soil property composition, other types of data, particularly microbiome composition, were much more susceptible. Microbiome quantification has been shifting from using OTUs to amplicon sequence variants (ASVs) which are designed to identify and retain more specific bacterial identification. Some microbes were overtly sparse across the samples and principal component regression could not successfully estimate model parameters. The microbiome table generation pipeline used for this field allows for the identification of very sparse microbes by way of using 99.5% identity clustering which results in 23,617 OTUs detected. After applying principal component regression, approximately 25% of the microbes were successfully modeled and retained. The methods proposed here should also be applicable to those types of tables as well with one caveat: ASV tables are sparser than OTU tables. While the methods here are zero-inflated, it is likely the percent of ASVs not successfully modeled would be larger than if using classical OTU techniques.

In conclusion, here we demonstrate the impact of spatially distributed soil property variations on several phenotypes of interest and present principal component regression as a method to alleviate the effects analytically. We observed that the microbiome communities were heavily influenced by the soil properties while the plant phenotypes were more resilient but nonetheless affected. Identifying

sources of variation and removing their influence enhances the ability to resolve other effects of interest and enables more honest, reliable, and believable quantification of subtle phenotypes.

## Materials and methods

In a related research manuscript (*Qi et al., 2022*), we describe analysis of the field dataset and include additional details on experimental design (sorghum varieties, watering regimes, and field layout). Microbes were not intentionally added to these experiments, but instead were allowed to evolve naturally from the environment. For clarity, we summarize key details here, as well. In short, we planted 24 varieties of sorghum in a split-plot design with 8 replicate blocks with 2 watering treatments per block (WW: well-watered and WS: drought). End of season harvesting procedures, microbiome sampling, DNA extraction, and sequencing are also fully described in *Qi et al., 2022*.

### Processing amplicon reads with VSEARCH and OTU table QC

Three microbiomes were collected for each plant: root endophytes, rhizosphere, and bulk soil, and all samples were sent for 16S PE amplicon sequencing at JGI (see *Qi et al., 2022* for extraction and sequencing methods). What follows is the VSEARCH (v2.9.0) (*Rognes et al., 2016*) workflow for taking the reads for each sample and processing them to curate the OTU table: merge paired ends, merge all samples, fastq filter, sequence dereplication, cluster unique sequences, remove chimeras, and read quantification. Merging paired ends had the following parameters: max diffs = 10, max diff percentage = 90, min merge length = 230, max merge length = 540. Samples were then combined into a single fasta file. Fastq filtering had the following parameters: maxee = 1, strip left = 19, strip right = 20, fastq max n's=0, fasta width = 0. Dereplication had the following parameters: min unique size = 1, fasta width = 0. OTU clustering had the following parameters: id percentage = 0.995, strand = both. Removing chimeras had the following parameters: fasta width = 0. Read quantification had the following parameters: id percentage = 0.9. These steps were combined all together in a directed acyclic graph (DAG) workflow and executed on a HTCondor high-throughput computation cluster. A total of 171,273 OTUs were detected and of those 114,179 had quantification across all 1280 samples. OTU table quality control was done in two steps: samples were removed if the total number of reads quantified across all OTUs was less than 10,000. In addition, OTUs were removed if the total number of reads quantified across all samples was less than 100 or greater than 200,000. After applying this filter 92,385 OTUs and 1280 samples remained. Of the OTUs removed only 422 had counts larger than 200,000 indicating the majority of the OTUs removed were rare and would not have enough information to perform proper statistical analysis. Once OTUs and samples that did not meet the quality control filters were removed, each OTU count in a sample was scaled proportionally to the same number, max number of reads per sample, so that all samples had the same number of OTU counts quantified.

### Geospatial interpolation methods

One major assumption of fitting spatial models is that the distribution is stationary, meaning that the mean and covariance between any two samples are the same throughout the grid. However, this field trial included two treatment factors (watering treatments and sorghum genotypes) which may have directly affected the measured soil properties. Thus, to satisfy the stationary assumption, we needed to account for any influence on the soil properties from the two treatment factors and/or their interaction. To do this, we fitted linear models including treatment factors and their interaction for each soil property as follows:

$$Property = intercept + Treatment + Genotype + Treatment * Genotype + residual\ error \qquad (1)$$

Then we calculated residuals for each property to generate stationary soil properties by subtracting the observed values by the predicted values based on model (1), as follows:

$$Residuals = Property_{ij} - Predicted\left(Property_{ij}\right) \qquad (2)$$

The residuals were used to estimate variances at multiple distances to produce a variogram and then, several spatial models were fitted to the data. The spatial models that were considered include (1) No Structure, (2) Exponential, (3) Spherical, (4) Gaussian, (5) Matern, and (6) Stein.

This was fully implemented in the R package 'gstat' using the fit.variogram() function, and full spatial model parameterizations can be found (*Pebesma and Wesseling, 1998*; *Pebesma, 2004*; *Pebesma and Wesseling, 1998*; *Pebesma, 2004*). The best fitting model was selected as the one that produced the minimal sum-of-squares errors. Given a spatial model, there are three metrics that describe a spatial fit and help us understand how the sampled points are correlated. The nugget is the estimated variance between two adjacent samples and represents the noise of the data. The range is the distance at which the change in variance with respect to distance first becomes zero and represents how far away sampled points demonstrate the correlation structure. Finally, the partial sill is the variance at the range minus the nugget variance. Since each spatial model is fully parameterized by only these three coefficients, model complexity is not a factor in estimating the model sum of squares to identify the best fitting parameterization.

With the best spatial model, we estimated spatial weights as a function of the distance between two sampling points and used these weights to predict values at nonsampled positions by kriging, a method to interpolate by using a weighted average of the observed values in the neighborhood of nonsampled position. More specifically, we applied ordinary kriging and let the sum of spatial weights to be one so that ordinary kriging is unbiased (*Olea, 2018*).

$$Residual_{krig}\left(position_0\right) = \sum_{i=1}^{n} \lambda_i Residual\left(position_i\right) \tag{3}$$

$$\text{where } \sum_{i=1}^{n} \lambda_i = 1$$

Kriging was performed using the krige() function in the R package 'gstat'.

## Statistical testing for evidence of spatial structure

Using the residuals of the soil properties from *Equation 2* (see Methods: Geospatial interpolation methods), multiple models were created with different spatial-covariance structures: intercept-only (no structure), spherical, exponential, Gaussian, linear, and rational-quadratic. A soil property is considered to exhibit evidence of spatial structure if any of five spatial covariance structure models are significantly more likely than the intercept-only model using likelihood ratio test (p value <0.05).

## Statistical testing for phenotype–property associations

Using only the soil properties that were determined to have significant spatial structure (see Methods: Statistical testing for evidence of spatial structure), the kriged soil property (*Equation 3*) was computed and used in the following association model for the mean structure:

$$f\left(Phenotype_{ijk}\right) = intercept + Treatment + Genotype + Treatment * Genotype + Property_{krig} + Z + error \tag{4}$$

where f() is the transformation function, *Z* is the multivariate normal spatial distribution nested within each split-plot replicate. In the case of plant height, fresh weight, and $\delta^{13}$C, the f() is the identity function and significant association is determined by computing a chi-square statistic based on type III sum of squares for the Property_krig. In the metabolite and microbiome estimation, f() is a Canberra distance transformation and significant association is determined by using constrained analysis of principal coordinates using the capscale() function in the vegan R package and constraining on all terms except Property_krig, which has its effect on metabolite composition estimated using permutation ANOVA with 999 iterations.

## Principal component regression

Starting with the stationary soil properties, that is, the residuals obtained from *Equation 2*, principal component analysis was performed using the PCA() function in the FactoMineR R package, and the first three components were extracted. Each PC was then kriged using the same methods described previously in the method section of geospatial interpolation methods using *Equation 3*. The kriged PCs were then used to create the following principal component regression model:

$$g\left(E\left(Phenotype\right)\right) \;=\; \alpha + \beta_1 PC1_{krig} + \beta_2 PC2_{krig} + \beta_3 PC3_{krig} + Z \qquad (5)$$

where g() is the link function for the expected value of the phenotype, $\alpha$ is the model intercept, each $\beta$ is the regression coefficient for the respective term, $Z$ is the multivariate normal spatial distribution nested within each split-plot replicate. In the case of plant height, fresh weight, and $\delta^{13}C$, the g() is the identity function. For the microbiome data, each tissue compartment was performed independently with *Equation 5*, and g() is the log link to the zero-inflated negative-binomial distribution implemented in the R package NBZIMM R package with the glmm.zinb() function. The metabolites were not modeled.

Soil property invariant data were created for the univariate phenotypes by extracting model residuals similar to *Equation 2*. For each microbiome compartment, every OTU was modeled independently and an adjusted count-like value was created by dividing the original count by the exponentiated regression coefficients for each PC as follows:

$$Adjusted\ OTU_i \;=\; Observed\ OTU_i\ /\ exp\left(b_1 + b_2 + b_3\right) \qquad (6)$$

where each $b$ coefficient is an estimated beta coefficient by fitting model (5).

## Soil property composition sampling and processing

Selected plants were excavated using a shovel to a depth of 12–14 inches. The soil (approximately 200 g) from the excavated root ball was shaken off into a wash pan in the field, homogenized and collected into a quart-size Ziploc bag. In addition to the collection of roots for microbiome analysis a subset of roots were collected for metabolite analysis as described in *Sheflin et al., 2019*. The soil used for chemical and physical analysis was stored in the Ziploc bags at 4°C and sent to Ward labs for analysis of pH, buffer pH, sum of cations , base saturation (%), soluble salts, organic matter, nitrate–nitrogen, phosphorus, potassium, calcium, magnesium, sodium, sulfur, zinc, iron, manganese, and copper.

## Root metabolomics sampling and processing
### Nontargeted metabolite profiling using GC–MS

Metabolite extraction was conducted by weighing out 19–21 mg of each freeze-dried sorghum root and placing them into clean 2 ml autosampler glass vials (VWR, Radnor, PA, USA). Automated control of sample extraction (i.e., solvent proportions, solvent volumes, sample agitation, and supernatant transfers) was accomplished using a standalone Gerstel MultiPurpose Sampler (MPS). Samples were extracted by adding 770 µl of methyl-tert-butyl-ether (MTBE) and 385 µl to each vial and vortexing on the MPS at room temperature for 30 min. To separate organic and aqueous layers, 640 µl of water was added to the remaining extract and vortexed for 15 min. Samples were then centrifuged for 25 min at 3500 rpm at 4°C. The organic layer was extracted twice by transferring into a new 2 ml autosampler vial without disturbing the lower layer then adding 600 µl of MTBE and transferring again. The aqueous layer was also extracted twice by transferring out of the vial into a new 2 ml autosampler vial without disturbing the pellet then adding 300 µl of methanol and 300 µl of acetonitrile, vortexing for 3 min and transferring again. The aqueous layer was completely dried under N gas, resuspended in 300 µl of 75% methanol. 20 µl of the aqueous layer from each sample was transferred to another set of glass vials, centrifuged for 2 min at 3500 rpm and then dried under $N_2$ (g) for 30 min. Dried samples were stored at −80°C until derivatization. Derivatization (methoximation and silylation) took place immediately prior to running the samples. Dried down samples were allowed to warm to room temperature and then resuspended in 50 µl of methoxyamine HCl (prewarmed to 60°C) and centrifuged for 30 s. Samples were then incubated at 60°C for 45 min, followed by a brief vortex, sonication for 10 min and an additional incubation at 60°C for 45 min. Following this, the samples were centrifuged before receiving 50 µl of *N*-methyl-*N*-(trimethylsilyl)trifluoroacetamide (MSTFA) + 1% trimethylchlorosilane (TMCS) (Thermo Fisher Scientific, Waltham, MA, USA), briefly vortexed and incubated at 60°C for 40 min, as described previously (*Chaparro et al., 2018*). Samples were loaded (~100 µl) into glass inserts within glass autosampler vials and centrifuged for 30 s prior to GC–MS analysis. In addition, a pooled extract was created by combining equal volumes of each sample into one glass vial for use as a consistent representative quality control sample (QC).

GC–MS analysis was performed using a Trace 1310 GC coupled to a Thermo ISQ mass spectrometer (Thermo Scientific). Derivatized samples (1 µl) were injected in a 1:10 split ratio. Metabolites were separated with a 30 m TG-5MS column (Thermo Scientific, 0.25 mm i.d. 0.25 µm film thickness). The GC program began at 80°C for 0.5 min and ramped to 330°C at a rate of 15°C per min and ended with an 8 min hold at a 1.2 ml min$^{-1}$ helium gas flow rate. The inlet temperature was held at 285°C and the transfer line was held at 260°C. Masses between 50 and 650 *m/z* were scanned at 5 scans/s after electron impact ionization.

Metabolomic data processing was conducted as previously described (*Chaparro et al., 2018*). GC–MS files were converted to.cdf format and processed by XCMS in R (*Smith et al., 2006*; *Mahieu et al., 2016*; *R Core Team, 2015*). All samples were normalized to the total ion current. RAMClustR was used to deconvolute peaks into spectral clusters for metabolite annotation (*Broeckling et al., 2014*). RAMSearch (*Broeckling et al., 2016*) was used to match metabolites using retention time, retention index, and matching mass spectra data with external databases including Golm Metabolome Database (*Hummel et al., 2007*; *Hummel et al., 2013*) and NIST (*Broeckling et al., 2016*).

## Leaf traits analyses

The middle portion (10–12 cm long) of the uppermost fully expanded leaf from individual plants was harvested in a coin envelope for the analysis of specific leaf area, C and N content, and stable isotopes of C and N. Leaf samples were oven-dried at 65°C to a constant mass and ca. 2.5 mg of the dry leaf was subsample using a custom-made leaf punch system in a tin capsule. The leaf punch provided the leaf area of subsample, which were weighed to estimate specific leaf area. The N, C and $\delta^{15}$N and $\delta^{13}$C concentrations of dry leaf were determined by combusting encapsulated samples in an elemental analyzer (ECS 4010, Costech Analytical Technologies) coupled to a continuous flow isotope ratio mass spectrometer (Delta XP, Finnigan MAT) at the Stable Isotope Core Laboratory, Washington State University.

## Software used and data availability

All analyses herein were performed in R using the following packages: raster(3.4.5), ggplot2(3.3.3), deldir(0.1.21), vegan(2.5.5), plyr(1.8.4), gridExtra(2.3), reshape(1.4.3), FactoMineR(2.4), factoextra(1.0.7), chngpt(2019.3.12), stringr(1.4.0), gstat(2.0.6), sp(1.4.2), scales(1.0.0), lme4(1.1.21), nlme(3.1.140), parallel(3.5.2), and patchwork(1.1.1). All data and scripts used to create all figures and perform all analyses can be found at https://zenodo.org/record/4715924.

## Acknowledgements

Funding: This work was supported by the US Department of Energy award DE_SC0014395.

## Additional information

### Competing interests

Rebecca S Bart: Reviewing editor, eLife. The other authors declare that no competing interests exist.

### Funding

| Funder | Grant reference number | Author |
|---|---|---|
| U.S. Department of Energy | DE_SC0014395 | Jeffrey C Berry<br>Mingsheng Qi<br>Balasaheb V Sonawane<br>Amy Sheflin<br>Asaph Cousins<br>Jessica Prenni<br>Daniel P Schachtman<br>Peng Liu<br>Rebecca S Bart |

The funders had no role in study design, data collection, and interpretation, or the decision to submit the work for publication.

### Author contributions

Jeffrey C Berry, Conceptualization, Data curation, Formal analysis, Methodology, Validation, Visualization, Writing – original draft, Writing – review and editing; Mingsheng Qi, Formal analysis, Investigation, Writing – review and editing; Balasaheb V Sonawane, Resources, Writing – review and editing; Amy Sheflin, Data curation, Resources; Asaph Cousins, Funding acquisition, Resources, Writing – review and editing; Jessica Prenni, Data curation, Resources, Supervision; Daniel P Schachtman, Conceptualization, Data curation, Funding acquisition, Methodology, Project administration, Resources, Supervision, Writing – review and editing; Peng Liu, Conceptualization, Formal analysis, Investigation, Methodology, Supervision, Writing – review and editing; Rebecca S Bart, Conceptualization, Funding acquisition, Project administration, Supervision, Writing – original draft, Writing – review and editing

### Author ORCIDs

Mingsheng Qi ![ORCID] http://orcid.org/0000-0003-2448-1227
Daniel P Schachtman ![ORCID] http://orcid.org/0000-0003-1807-4369
Rebecca S Bart ![ORCID] http://orcid.org/0000-0003-1378-3481

### Decision letter and Author response

Decision letter https://doi.org/10.7554/eLife.70056.sa1
Author response https://doi.org/10.7554/eLife.70056.sa2

## Additional files

### Supplementary files

• Transparent reporting form

### Data availability

The 16S microbiome sequencing was done at JGI and was previously published (https://doi.org/10.1101/2021.04.13.437608). Raw sequence data are available for download through the JGI user portal: Author: Daniel Schachtman; Title: "Systems Analysis of the Physiological and Molecular Mechanisms of Sorghum Nitrogen Use Efficiency, Water Use Efficiency and Interactions with the Soil Microbiome" at the following url: https://genome.jgi.doe.gov/portal/pages/projectStatus.jsf?db=EneSor2017_itags_14. Data are listed under the following title: Energy Sorghum Plate 2017_"62-98" itags. Alternatively, all raw data are available for download from the project website: http://shiny.danforthcenter.org/sorghum_systems/. Additional data and code associated with this manuscript are available here: https://doi.org/10.5281/zenodo.4715924.

The following dataset was generated:

| Author(s) | Year | Dataset title | Dataset URL | Database and Identifier |
|---|---|---|---|---|
| Berry JC | 2021 | Increased signal to noise ratios within experimental field trials by regressing spatially distributed soil properties as principal components | https://zenodo.org/record/4715924#.YlaYmtPMLs0 | Zenodo, 10.5281/zenodo.4715924 |

The following previously published datasets were used:

| Author(s) | Year | Dataset title | Dataset URL | Database and Identifier |
|---|---|---|---|---|
| Qi M | 2021 | Interacting beneficial and detrimental responses to drought in the sorghum microbiome | https://www.ncbi.nlm.nih.gov/bioproject/PRJNA720397 | NCBI BioProject, PRJNA720397 |

*Continued on next page*

*Continued*

| Author(s) | Year | Dataset title | Dataset URL | Database and Identifier |
|---|---|---|---|---|
| Schachtman D | 2021 | Systems Analysis of the Physiological and Molecular Mechanisms of Sorghum Nitrogen Use Efficiency, Water Use Efficiency and Interactions with the Soil Microbiome | https://genome.jgi.doe.gov/portal/pages/projectStatus.jsf?db=EneSor2017_itags_14 | JGI, 1191873 |

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
