## [Editor Report]

In this study, the authors took an experimental, empirical approach to tackle the thorny problem of micro-scale variation in soil properties within and among field plots in confounding statistical analyses. The issue is that in field experiments, small variation in one or more soil property variables can obscure true effects of experimental variables on plant phenotypes. The main result is that without their framework they would not have found the association between water treatment, plant growth and Microvirga bacterial abundance, it would have been lost to the noise inherent in these kind of large-scale experiments with relatively modest degrees of freedom. Overall, the PC-based approach to de-noise these kinds of datasets provides an important advance by pulling out subtle phenotypic effects in field trials.

---

## [Decision Letter]

**Decision letter after peer review:**

Thank you for submitting your article "Increased signal to noise ratios within experimental field trials by regressing spatially distributed soil properties as principal components." for consideration by *eLife*. Your article has now been reviewed by 3 peer reviewers, one of whom is a member of our Board of Reviewing Editors, and the evaluation has been overseen by Meredith Schuman as the Senior Editor. The following individual involved in review of your submission has agreed to reveal their identity: Andrew Gloss (Reviewer #2).

Essential revisions:

1) Each of the reviewers' specific concerns below should be addressed prior to resubmission of a revised manuscript for publication consideration. A major concern was a lack of detail on many aspects of the statistical model, the location of the raw data and ability of others to re-run the code. This should be the focus of the author's should they consider revising and resubmitting this manuscript.

2) Because the manuscript is, essentially a methods development report, it is of the utmost importance that the authors revise this manuscript in alignment with those goals. It should be straightforward, not difficult, for others to implement the tools developed.

*Reviewer #1 (Recommendations for the authors):*

I would like to see the raw data and methods for all of these experiments uploaded/appended as supplementary information here--it isn't satisfactory that one should turn to a preprint for this information--it should all be in the present manuscript. In that vein, I was also confused by whether microbes were a treatment or a passive variable here in this study (e.g., I had a hard time differentiating between the Qi et al., biorxiv manuscript and this one).

*Reviewer #2 (Recommendations for the authors):*

Overall, I found the approach exciting, and the paper makes a good case for its use in future experiments! However, some work is needed to make the presentation more complete in a few different ways, as numbered below. When revising, please consider the following sentence-by-sentence and analysis-by-analysis: is the approach fully described in the methods? Are the necessary results provided so a reader could evaluate support for the statement rather than taking the written interpretation at face value? (e.g., L245: readers should be able to view the results, rather than relying on the vague assurance that the results that aren't shown are "similar" to the one that is. This paper could and probably should have more extensive supplementary results to achieve this).

1. Not all aspects of the method's implementation are described thoroughly in the paper, in both practical and conceptual senses. This includes both the specific formulations of each analysis -- with a careful description of the model inputs, outputs, specification, and software packages -- and very clear explanations of the goal of each step and how the approach achieves it. The incompleteness of these details made it difficult to fully evaluate, and could pose obstacles to its reliable implementation and interpretation by other researchers. In a paper presenting a methodological approach and arguing for its broader use, clearly walking readers through these steps is essential. Addressing this will require revisions to the methods and results alike. While I do appreciate the brevity, other papers implementing spatial models for environmental variation -- such as Pauli et al., 2018 (G3, doi: 10.1534/g3.117.300479) or Velazco et al., 2017 (Theor. Appl. Genet; doi: 10.1007/s00122-017-2894-4) -- walk the reader through their approaches more thoroughly.

2. Results are presented incompletely. For example, Figure 4E shows how PC regression affected variance partitioning among explanatory variables and unexplained noise, but only for one dependent variable. This really should be conducted for every dependent variable included in the study. When conducting many tests, a seemingly compelling result can arise even by chance, so it's difficult to know how to interpret a single strong result that is hand-picked to be presented. Similarly, the effect of spatial modeling should be presented across *all* OTUs considered in the study, not just one with a particularly strong effect in the desired direction. Otherwise, one is left wondering if this is a chance result (since re-fitting a model on adjusted data will always alter it) that only arose because so many OTUs were tested -- and whether patterns in the opposite direction, where a significant effect emerged only in the unadjusted phenotypes, also were observed. Note that multiple test corrections (likely FDR) should be presented when applicable as well throughout the paper, especially for the large number of OTU tests that must've been conducted but not shown. It seems like the results should pass this scrutiny, but it must be applied nonetheless!

3. How the approach builds on or is unique from previous studies and approaches, and ensuing strengths and weaknesses to be expected as a result, is not sufficiently described; see Public Review for further details.

Specific comments:

I don't quite follow how separation of cluster centers in the PCA plots (Figures 4A-B) suggests that environmental noise has been reduced and treatment signals have been boosted. In these figures, the centers for each condition do appear further apart, but the spread of points within each condition have also increased. If both spread and centers increase, does this actually reflect better separation of conditions, or just a re-scaling of the overall plot? Does variance partitioning (explained vs. unexplained variance) and the significance of the condition effect in a statistical model actually improve? (Also, I'm confused by L223-224 -- wouldn't weakening a true treatment effect also enlarge clusters, albeit in a different way by drawing points toward the center of the plot?)

It would be very helpful to walk through the spatial model selection more -- a bit on the different spatial covariance structures that were tested in particular. The test results used for model selection should also be more fully presented, including parameters relevant to the likelihood ratio tests that were conducted (e.g., degrees of freedom for each comparison, log likelihood scores for each model and associated p-value, etc).

L143-147: Microbiomes of different compartments being affected by different soil elements seems plausible, but this could also just reflect false negatives, a common problem when simply "tallying up" the tests significant at some threshold. As a result of experimental noise (or differences in the amount of unexplained variance within specific compartments), it's possible that an element might have significant effects only in one compartment even if it affects all of them. Evaluating a model of microbiome_composition ~ element * compartment is needed to test this, paying attention the significance of the interaction term.

Table S1: How is model complexity taken into account? Typically, a more complex model will always have reduced error. Do the model comparisons penalize for increasing model complexity?

I was unable to access the scripts on Zenodo, but maybe that was user error on my end!

*Reviewer #3 (Recommendations for the authors):*

1. It might be helpful to describe the details of the statistical model.

2. To help readers employ the proposed tool in their studies, it is valuable to include the step-by-step procedure of the proposed strategy, such as selection of variables, number of PCs to include in the model etc.

3. For the proposed tool, will the correlations between the phenotypes affect the results/performance?

4. Page 10, line 151: GCMS should be "GC/MS" or "GC-MS"?

---

## [Author Response]

Essential revisions:1) Each of the reviewers' specific concerns below should be addressed prior to resubmission of a revised manuscript for publication consideration. A major concern was a lack of detail on many aspects of the statistical model, the location of the raw data and ability of others to re-run the code. This should be the focus of the author's should they consider revising and resubmitting this manuscript.2) Because the manuscript is, essentially a methods development report, it is of the utmost importance that the authors revise this manuscript in alignment with those goals. It should be straightforward, not difficult, for others to implement the tools developed.

We agree completely and appreciate the editor and reviewer suggestions to help us achieve that goal. With this in mind, we have extensively revised the methods section and particularly appreciate Reviewer #2 pointing us towards examples of previous papers. In addition to revising the methods section, we have addressed the other points raised by each reviewer below.

Reviewer #1 (Recommendations for the authors):I would like to see the raw data and methods for all of these experiments uploaded/appended as supplementary information here--it isn't satisfactory that one should turn to a preprint for this information--it should all be in the present manuscript. In that vein, I was also confused by whether microbes were a treatment or a passive variable here in this study (e.g., I had a hard time differentiating between the Qi et al., biorxiv manuscript and this one).

We thank the reviewer for these comments and for the very nice summary of our work. We have revised this submission to include zenodo links to the raw data, scripts and methods, as requested (Line 529). In this paper, the microbiome is a passive variable (not an active treatment that we applied to the field). We have clarified this in line 343. We have also edited the text in many places to clarify the relationship between this methods paper, and the related research paper (Qi, M. et al., 2021). We are also happy to share that the Qi et al., manuscript was recently accepted at ISME J. As soon as we have an updated doi, we will update that in this manuscript, as well. While both papers use the field plant data and the microbiome data, the two papers each describe additional unique datasets and report different results.

Reviewer #2 (Recommendations for the authors):Overall, I found the approach exciting, and the paper makes a good case for its use in future experiments! However, some work is needed to make the presentation more complete in a few different ways, as numbered below. When revising, please consider the following sentence-by-sentence and analysis-by-analysis: is the approach fully described in the methods? Are the ecessaryy results provided so a reader could evaluate support for the statement rather than taking the written interpretation at face value? (e.g., L245: readers should be able to view the results, rather than relying on the vague assurance that the results that aren’t shown are “similar” to the one that is. This paper could and probably should have more extensive supplementary results to achieve this).

We thank the reviewer for the encouraging words, thorough evaluation of our work and many helpful suggestions for how to improve the manuscript. We have extensively revised the manuscript as described for each point below.

1. Not all aspects of the method’s implementation are described thoroughly in the paper, in both practical and conceptual senses. This includes both the specific formulations of each analysis – with a careful description of the model inputs, outputs, specification, and software packages – and very clear explanations of the goal of each step and how the approach achieves it. The incompleteness of these details made it difficult to fully evaluate, and could pose obstacles to its reliable implementation and interpretation by other researchers. In a paper presenting a methodological approach and arguing for its broader use, clearly walking readers through these steps is essential. Addressing this will require revisions to the methods and results alike. While I do appreciate the brevity, other papers implementing spatial models for environmental variation – such as Pauli et al., 2018 (G3, doi: 10.1534/g3.117.300479) or Velazco et al., 2017 (Theor. Appl. Genet; doi: 10.1007/s00122-017-2894-4) -- walk the reader through their approaches more thoroughly.

Thank you for this concern and also for the suggestions. We have extensively revised the methods sections to include full model parameterizations and descriptions. We also thank the reviewer for pointing out additional references that should be included. In addition to citing these previously described approaches in the intro, we have revised the discussion to include a ‘compare and contrast’ between these previous approaches and our method.

2. Results are presented incompletely. For example, Figure 4E shows how PC regression affected variance partitioning among explanatory variables and unexplained noise, but only for one dependent variable. This really should be conducted for every dependent variable included in the study. When conducting many tests, a seemingly compelling result can arise even by chance, so it's difficult to know how to interpret a single strong result that is hand-picked to be presented.

Thank you for the suggestion regarding adding more variance components plots to the manuscript. We have exchanged the principal components plot (Figure 4B) with variance components to show the effects of the principal component regression on the microbiome composition. The PCA plot is now moved to supplemental in addition to showing the same treatment effect in the other two compartments, soil and root (Figure 4—figure supplement 1). All harvest data is now included in Figure 4—figure supplement 2, as requested.

Similarly, the effect of spatial modeling should be presented across all OTUs considered in the study, not just one with a particularly strong effect in the desired direction. Otherwise, one is left wondering if this is a chance result (since re-fitting a model on adjusted data will always alter it) that only arose because so many OTUs were tested -- and whether patterns in the opposite direction, where a significant effect emerged only in the unadjusted phenotypes, also were observed. Note that multiple test corrections (likely FDR) should be presented when applicable as well throughout the paper, especially for the large number of OTU tests that must've been conducted but not shown. It seems like the results should pass this scrutiny, but it must be applied nonetheless!

We agree with the reviewer’s concern regarding representativeness of the results. Following your suggestion, we have conducted tests for all OTUs and added additional text to the manuscript reflecting the result with control of FDR (line 266). We also have added supplemental figures showing the results of the change-points before and after (Figure 4—figure supplement 4).

3. How the approach builds on or is unique from previous studies and approaches, and ensuing strengths and weaknesses to be expected as a result, is not sufficiently described; see Public Review for further details.

Again, we thank the reviewer for encouraging us to more fully place this work in the context of the field. We have revised both the introduction and the discussion with this goal in mind.

Specific comments:I don't quite follow how separation of cluster centers in the PCA plots (Figures 4A-B) suggests that environmental noise has been reduced and treatment signals have been boosted. In these figures, the centers for each condition do appear further apart, but the spread of points within each condition have also increased. If both spread and centers increase, does this actually reflect better separation of conditions, or just a re-scaling of the overall plot? Does variance partitioning (explained vs. unexplained variance) and the significance of the condition effect in a statistical model actually improve? (Also, I'm confused by L223-224 -- wouldn't weakening a true treatment effect also enlarge clusters, albeit in a different way by drawing points toward the center of the plot?)

Thank you for giving this part considerable thought. Since the microbiome counts are brought back to the original count space after doing the soil property adjustment, we feel that there are negligible re-scaling effects. For instance, we would not expect all counts to increase thus forcing the clusters further apart simply due to scale. OTUs are adjusted upwards in some samples and down in others. However, considering this and other comments from the reviewers, we decided to move previous panel 4B to supplemental in exchange for a variance component analysis on the compositions.

It would be very helpful to walk through the spatial model selection more -- a bit on the different spatial covariance structures that were tested in particular. The test results used for model selection should also be more fully presented, including parameters relevant to the likelihood ratio tests that were conducted (e.g., degrees of freedom for each comparison, log likelihood scores for each model and associated p-value, etc).

We leverage an existing R package, gstat, and all possible types of models (nugget, exponential, spherical, Gaussian, Matern, and Stein) were evaluated. Full parameterizations of these can be found in their manuscripts (Pebesma and Wesseling 1998; Pebesma 2004)(Pebesma 1998, Pebema 2004). These models are fully parameterized by the same three coefficients (nugget, sill, and range) and we have revised the text to clarify their explicit definitions (methods) as well as added in an additional citation. As suggested, we have extensively revised the methods section to more fully walk the reader through the process.

L143-147: Microbiomes of different compartments being affected by different soil elements seems plausible, but this could also just reflect false negatives, a common problem when simply "tallying up" the tests significant at some threshold. As a result of experimental noise (or differences in the amount of unexplained variance within specific compartments), it's possible that an element might have significant effects only in one compartment even if it affects all of them. Evaluating a model of microbiome_composition ~ element * compartment is needed to test this, paying attention the significance of the interaction term.

We appreciate the amount of thought that went into this suggestion and how interpretable the results are from a biological point of view. From a statistical standpoint, a model that includes a compartment interaction term allows different compartments react differently to the changing element values. As suggested, we have completed the analysis including an interaction term and have included an additional table that directly assesses the effect of the interaction term (Figure 2—figure supplement 1). We have also updated the text to include this analysis in the results. The statistical test to decide if interaction is significant is limiting in its interpretability. We therefore also performed post-hoc tests to investigate more directly which compartments are affected.

Table S1: How is model complexity taken into account? Typically, a more complex model will always have reduced error. Do the model comparisons penalize for increasing model complexity?

This is an absolutely accurate statement and something geospatial statistics has aimed to alleviate. For all the different types of spatial models considered here, each one of them is fully parameterized by the same three coefficients (nugget, sill, and range) so from a degrees of freedom and error point of view, they are all equally complex. We identify which spatial model is “best” by finding the model that produces the smallest error, and since the complexities are the same, this metric is suitable for this determination. We have added a sentence in the methods of the manuscript to bring more clarity to the errors of each type of model.

I was unable to access the scripts on Zenodo, but maybe that was user error on my end!

We have updated the link and verified that others can access the site (the previous link may have been locked!). Hopefully it now works for you, as well. If not, please just let *eLife* staff know and we’ll sort it out.

Reviewer #3 (Recommendations for the authors):1. It might be helpful to describe the details of the statistical model.

Thank you for this suggestion. We have extensively revised the methods sections, including specifying model equations, to give a more systematic look at exactly what was done. We also have included all scripts and raw data in a zenodo repository.

2. To help readers employ the proposed tool in their studies, it is valuable to include the step-by-step procedure of the proposed strategy, such as selection of variables, number of PCs to include in the model etc.

We think the use of the word ‘tool’ was misleading since we are not providing a deployable R package for general use. Therefore, we have changed this word to ‘method’. Regardless, we like this suggestion and so have updated the text in the Discussion section to highlight what we feel is the balancing act between the number of PC’s to use vs the number of degrees of freedom required to adjust. This is an important concept, thank you for the suggestion.

3. For the proposed tool, will the correlations between the phenotypes affect the results/performance?

Good question. In this manuscript we are only considering phenotypes one at a time to assess for soil property variations and removing those effects. However, in our related paper (Qi et al.,), after de-noising the data, we leveraged correlations between experimental systems and between plant phenotypes to further exclude potential ‘false positives.’

4. Page 10, line 151: GCMS should be "GC/MS" or "GC-MS"?

Corrected.